# Graphene-Based Tunable Dual-Frequency Terahertz Sensor

**DOI:** 10.3390/nano14040378

**Published:** 2024-02-18

**Authors:** Maixia Fu, Yuchao Ye, Yingying Niu, Shaoshuai Guo, Zhaoying Wang, Xueying Liu

**Affiliations:** 1Key Laboratory of Grain Information Processing and Control, Ministry of Education, Henan University of Technology, Zhengzhou 450001, China; ye15565219553@163.com (Y.Y.);; 2Henan Key Laboratory of Grain Photoelectric Detection and Control, Henan University of Technology, Zhengzhou 450001, China; 3College of Information Science and Engineering, Henan University of Technology, Zhengzhou 450001, China

**Keywords:** terahertz, graphene, sensor, dual-frequency, tunable

## Abstract

A tunable dual-band terahertz sensor based on graphene is proposed. The sensor consists of a metal bottom layer, a middle dielectric layer, and single-layer graphene patterned with four strips on the top. The numerical simulations results show that the proposed sensor exhibits two significant absorption peaks at 2.58 THz and 6.07 THz. The corresponding absorption rates are as high as nearly 100% and 98%, respectively. The corresponding quality factor (Q) value is 11.8 at 2.58 THz and 29.6 at 6.07 THz. By adjusting the external electric field or chemical doping of graphene, the positions of the dual-frequency resonance peak can be dynamically tuned. The excitation of plasma resonance in graphene can illustrate the mechanism of the sensor. To verify the practical application of the device, the terahertz response of different kinds and different thicknesses of the analyte is investigated and analyzed. A phenomenon of obvious frequency shifts of the two resonance peaks can be observed. Therefore, the proposed sensor has great potential applications in terahertz fields, such as material characterization, medical diagnosis, and environmental monitoring.

## 1. Introduction

The frequency of terahertz wave lies between infrared and microwave, which is just in the transition range from macroscopic electronics to microscopic photonics [1,2]. It has many excellent properties such as wide spectrum, coherence, nondestructive, and fingerprint absorption spectrum. Therefore, terahertz wave has important applications in communication, nondestructive testing, medicine, astronomy, and quantum key distribution [3,4,5,6,7]. However, there are many challenges in the development of terahertz technology, such as making stable and efficient on-chip terahertz generation, designing actively tunable terahertz modulators, improving the response speed and sensitivity of terahertz detectors, etc. In recent years, with the rapid development of photonics and nanotechnology, terahertz sources with high power and terahertz detectors with high sensitivity have made great achievements [8,9,10,11,12,13]. However, due to the lack of materials that can effectively interact with terahertz waves, research on terahertz modulating devices is still in its initial stage and yet to be developed.

Metamaterials, having artificial periodic structures and exhibiting extraordinary physical properties [14,15], such as inverse Doppler effect, negative refractive index, and negative magnetic permeability, have been widely used in super lenses [16], cloaking [17], filters [18], and perfect absorbers [19,20]. Most of the early metamaterials were made of metal, and once the structural parameters are designed, the resonant frequencies are difficult to change again. This means that terahertz waves can only be passively regulated. Therefore, it is necessary and meaningful to design dynamically tunable terahertz devices. Graphene, as a two-dimensional photoelectric material, has been regarded as the most promising material for active terahertz modulation because its optical conductivity in the terahertz frequency range can be changed by the external bias voltage or chemical doping. So far, there have been many reports on graphene-based metamaterials, such as graphene-based modulators [21], graphene-based absorbers [22], graphene-based sensors [23], graphene-based hyperbolic metamaterial [24,25], etc. Among these applications, terahertz sensors have become a research focus due to their important applications in biomedical, food safety, chemical analysis, and environmental monitoring. For example, in 2017, Wang et al. [26] proposed an ultra-narrow terahertz perfect absorber to fulfill refractive index sensing, which is constructed with a square resonator. The absorption rate was 98.86%, the bandwidth was 0.02 THz, the sensitivity was 2.58 THz/RIU (Refractive Index Unit, RIU), the Q value was 442, and the FOM value was 385.07. In 2020, Hu et al. [27] proposed a terahertz refractive index sensor based on a perfect metamaterial absorber (PMA). The sensor consists of an array of cross-shaped resonators on a copper/silicon substrate, and a near-perfect absorption peak was obtained at 2.44 THz. The corresponding sensitivity, Q value, and FOM value could reach 1.94 THz/RIU, 637, and 506 RIU^−1^, respectively. In 2021, Chen et al. [28] proposed a terahertz sensor with a dual-band tunable absorption characteristic based on graphene surface plasmon resonance (SPR). The two ultra-narrow perfect absorption peaks were at 3.74 THz and 7.73 THz, respectively. The sensor was dynamically adjustable and polarization-insensitive. The sensor had the maximum sensitivity of 2.475 THz/RIU, Q value of 216.29, and FOM value of 76.89 RIU^−1^. In 2022, Nickpay et al. [29] proposed a graphene-based tunable three-band sensor which was composed of a graphene split ring resonator and a patch resonator. Three absorption peaks were at 4.66 THz, 6.035 THz, and 8.72 THz, with absorption rates of 99.2%, 99%, and 89%, respectively. The SPR in graphene was responsible for the mechanism. The sensitivity was 1791 GHz/RIU, the Q value was 32.354, and the FOM value was 7.046 RIU^−1^. In the same year, Li et al. [30] proposed a four-band terahertz sensor based on Fano resonance. The maximum sensitivity, Q value, and FOM value of four transmission peaks were up to 200 GHz/RIU, 177, and 26.7 RIU^−1^, respectively. In addition, the sensor was polarization-insensitive in a wide angle from 0° to 50°, which has potential applications in biomedical diagnostics and environmental monitoring.

Although great progress has been made in the research of terahertz sensors and broad application prospects have been provided in various fields, there are still many challenges due to limitations in technological level and manufacturing processes. For example, the sensitivity and response speed of terahertz sensors need to be further improved to meet the practical application needs. In addition, compared to traditional single-frequency sensing, dual-frequency or multi-frequency sensing often introduces multiple resonance modes in structural design. The tunability of the devices has indeed increased, but complex structures provide difficulties for production and manufacturing. In this paper, a dual-frequency terahertz sensor based on patterned graphene is proposed, which has a relatively simple structure, high sensitivity, and active tunability.

## 2. Design and Mechanism

The schematic diagram of the graphene-based dual-frequency terahertz sensor is shown in Figure 1a. The sensor is a classical sandwich structure consisting of a single-layer patterned graphene on the top, an intermediate dielectric layer, and a metal substrate at the bottom. Graphene is designed as an array of four strips to enhance the coupling effect with incident light and excite plasma resonance. It is known that plasma is very sensitive to external electromagnetic fields. When the external dielectric environment changes, the resonance peak will give a frequency shift, thereby achieving a sensing effect. The intermediate layer is designed to isolate the direct interaction between the gold substrate and graphene. Meanwhile, an appropriate intermediate material can adjust the impedance matching between the incident wave and the metamaterial, further enhancing the absorption of the incident wave. It has been reported that TOPAS can maintain a stable dielectric constant and has lower absorption losses in the terahertz frequency band. Therefore, a copolymer of cyclic olefins (TOPAS) with a relative dielectric constant of 2.35 and a thickness of 11 μm is chosen as the intermedium in our work. Gold film serves as a substrate to prevent the transmission of terahertz waves and can also provide support for the device. Figure 1b is the top view of the unit cell. Figure 1c depicts the geometric parameters of the unit cell. The four strips can be divided into two parallel groups, and the corresponding lengths and widths are marked as L_1_, d_1_, L_2_, and d_2_. The periodicity in x and y directions is the same and labeled as p.

A surface conductivity model is employed to numerically simulate monolayer graphene, and the surface conductivity *σ*(*ω*) can be calculated by using the Kubo formula [31]:$$\sigma (\omega ,\mu_{c},\tau ,T)=\sigma_{\mathrm{intra}\;}(\omega ,\mu_{c},\tau ,T)+\sigma_{\mathrm{inter}\;}(\omega ,\mu_{c},\tau ,T)$$
(1)$$\begin{array}{c} \sigma_{\mathrm{intra}}(\omega )=\frac{i}{\omega +i/\tau }\frac{2e^{2}k_{B}T}{\pi \hbar^{2}}\ln[2\cosh(\frac{\mu_{c}}{2k_{B}T})]\\ \sigma_{\mathrm{inter}}(\omega )=\frac{e^{2}}{4\hbar^{2}}[\frac{1}{2}+\frac{1}{\pi }\arctan(\frac{\hbar \omega -2\mu_{c}}{2k_{B}T})-\frac{i}{2\pi }\ln\frac{{\left(\hbar \omega +2\mu_{c}\right)}^{2}}{{\left(\hbar \omega -2\mu_{c}\right)}^{2}+4{\left(k_{B}T\right)}^{2}}] \end{array}$$
where *σ_inter_* comes from the contribution of interband transitions, and *σ_intra_* comes from the contribution of intraband transitions. *ω* is the angular frequency of the electromagnetic wave, *k_B_* is the Boltzmann constant, *ħ* = h/2π is the reduced Planck constant, and *T* is the absolute temperature at 300 K. *e* is the charge of an electron. *τ* is the relaxation rate and can be calculated as $\tau =\mu \mu_{c}/{eV}_{f}^{2}$, where μ is the electron mobility and $V_{f}$ is the Fermi velocity. The Fermi level of graphene is labeled as *μ_c_*. In the terahertz frequency domain, the photon energy satisfies *ħω* << *μ_c_* and *μ_c_* >> *k_B_T*; the photon energy is much smaller than the Fermi level. Therefore, Formula (1) can be simplified to a quasi-Drude model [31]:(2)$$\sigma_{g}\approx \sigma_{intra}(\omega )=\frac{e^{2}\mu_{c}}{\pi \hbar^{2}}\frac{i}{\left(\omega +i/\tau \right)}$$

The variation in the trends of the real and imaginary parts of $\sigma_{g}$ under a different Fermi level *μ_c_* is investigated and shown in Figure 2a,b. It can be observed that the real part of $\sigma_{g}$ decreases with the frequency increasing, which can be attributed to the electronic response of graphene. The imaginary part initially increases and then decreases with frequency, which is caused by the energy dissipation of graphene. When the operating frequency is fixed, both the real and imaginary parts of the conductivity increase with the increase in Fermi level, and the density of the charge carriers in graphene is responsible for this phenomenon.

The proposed sensor is studied by using Lumerical FDTD software (version 2019, Canonsburg, PA, USA). Figure 3 shows the simulation model established in FDTD. The plane wave is selected as the incident wave, the periodic boundary conditions are set in the x-axis and y-axis directions, and the z-direction boundary condition is set as the perfect matching layer (PML); the adaptive mesh and local mesh refinement are used to improve the simulation accuracy. Different monitors are added to the simulation model to collect the data of transmission, reflection, and field intensity distribution. By calculating the reflection coefficient S_11_ and the transmittance coefficient S_21_, the absorption rate can be calculated as A(ω) = 1 − R(ω) − T(ω) = 1 − |S_11_|^2^ − |S_21_|^2^, where R(ω) and T(ω) are reflectance and transmittance, respectively.

## 3. Results and Discussion

The impact of the geometric parameters of graphene pattern on sensing performance is studied and analyzed, while the Fermi level of graphene is set to be 0.8 eV. Meanwhile, in order to facilitate the description of dual-frequency peaks, the low-frequency resonance peak at 2.58 THz is labeled as P_1_ and the high-frequency resonance peak at 6.07 THz is labeled as P_2_.

Figure 4a shows the effect of L_1_ on P_1_ and P_2_ in the condition of transverse magnetic (TM) polarization when L_2_ = 3.7 μm, d_1_ = 1 μm, and d_2_ = 1.5 μm. It can be observed that as L_1_ gradually increases from 3 µm to 8 µm, there is an apparent redshift for P_1_, while P_2_ remains nearly unchanged. The bandwidth of P_1_ narrows as L_1_ increases. Figure 4b shows the effect of d_1_ on P_1_ and P_2_ when L_1_ = 7 μm. As d_1_ increases from 1 µm to 5 µm, both the amplitude and position of P_1_ and P_2_ are affected. For P_1_, the absorption rate significantly decreases, the position of peak shifts to a higher frequency, and the bandwidth widens. For P_2_, the absorption rate also decreases and the bandwidth widens, but the position of the peak shifts to a lower frequency. The influence of the length (L_2_) and width (d_2_) of the graphene strips parallel to the y-axis on P_1_ and P_2_ is studied and shown in Figure 4c,d when L_1_ = 7 μm and d_1_ = 1 μm. It is clear that the changes in d_2_ and L_2_ have a relatively weak impact on P_1_. For P_2_, as L_2_ increases, the absorption rate slightly decreases, and the bandwidth slightly widens. However, as d_2_ increases, the P_2_ peak shows a significant blue shift, and the bandwidth widens. These results illustrate that P_1_ is generated mainly due to the interaction between two lateral graphene strips and terahertz, while P_2_ is due to the longitudinal graphene strips and terahertz.

To investigate the resonance mechanism of the proposed sensor, the distribution of the electric field and surface currents on the graphene surface at the condition of a normal incidence angle is analyzed and depicted in Figure 5 and Figure 6. Figure 5a,b depict the electric field at resonance frequencies of 2.58 THz (P_1_) and 6.07 THz (P_2_) under TM polarized excitation in the x-y plane. Figure 5c,d depict the electric field in the x-z plane, respectively. As shown in Figure 5a,c at the resonance frequency of 2.58 THz (P_1_), the electric field is mainly concentrated at both ends of the graphene strips parallel to the x-axis, exhibiting characteristics of localized plasmon excitations. As shown in Figure 6a, the surface current generated at 2.58 THz is mainly concentrated on the graphene strips parallel to the y-axis, and the current propagates in the direction of the x-axis. The aggregation of positive and negative charges forms a strong electric dipole resonance. Similarly, at a resonance frequency of 6.07 THz, as shown in Figure 5b,d, the electric field is mainly concentrated at the edge of the graphene strips parallel to the y-axis, forming an electric dipole resonance. The corresponding surface current, as shown in Figure 6b, is mainly concentrated on graphene strips parallel to the y-axis and propagates along the direction of the x-axis.

The unique tunable properties of graphene can be achieved by applying a bias voltage to change its Fermi level, which can affect the carrier concentration and achieve dynamic tunability of the sensor. The relationship between the Fermi level *μ_c_* and the bias voltage *V_g_* can be described as follows [32]:(3)$$|u_{c}|=\hbar V_{f}\sqrt{\pi |{a_{0}V}_{g}|}$$
where $a_{0}=\frac{\varepsilon_{0}\varepsilon_{h}}{ed}$ is the capacitance of the structural model, $\varepsilon_{0}$ is the vacuum intermediate constant, $\varepsilon_{h}$ is the dielectric constant, and *d* is the thickness of the middle dielectric layer. As shown in Figure 7, the structure parameters are set as L_1_ = 7 μm, L_2_ = 3.7μm, d_1_ = 1 μm, and d_2_ = 1.5 μm; when the Fermi level (μ_c_) increases from 0.4 eV to 0.8 eV, both of the dual-frequency absorption peaks give remarkably blue shift and bandwidth widening. P_1_ moves from 1.905 THz to 2.857 THz, while P_2_ moves from 4.444 THz to 6.805 THz. The reason for this is an enhancement of plasma resonance, which originated from the increasing in the density of the charge carriers in graphene. The wave vector of the plasma polarization exaction in graphene can be described as $k_{spp}\propto hf_{r}^{2}/(2\alpha_{0}E_{f}c)$, where fr is the resonant frequency and $\alpha_{0}$ is the fine structure constant. The frequency of graphene plasmon resonance is related to the Fermi level. When μ_c_ reaches 0.7 eV, plasmon resonance reaches its maximum. As the Fermi level continues to increase, the frequency of the absorption peak gradually deviates from the plasmon resonance region, and the amplitude of the peak begins to decrease.

The sensitivity of TE (transverse electric) and TM (transverse magnetic) polarization for the designed sensor is also investigated and shown in Figure 8. It is clear that the two different polarization modes have a significant impact on the absorption peak. On the one hand, the asymmetry of the designed structure makes the sensor sensitive to the polarization angle. On the other hand, the dual-frequency resonance peaks are affected by different strips. For four strips in a single unit cell, two are parallel to the x-axis and two are parallel to the y-axis. When interacting with incident light, the local plasma resonance excited is different, thus leading to differences in the absorption peak.

To characterize the application performance of the proposed device, a layer of analyte is considered on the top of the sensor. The effect of the analyte’s thickness on sensor performance is studied and shown in Figure 9a. Here, frequency shift (FS) is defined as FS = *f*_t_ − *f*_r_, where *f*_t_ is the resonant frequency when covering a layer of analyte with a thickness of t, and *f*_r_ is the resonant frequency when there is no analyte. In Figure 9a, the analyte’s refractive index is assumed to be 1.5; both of the P_1_ and P_2_ tend to show as redshifted as the thickness increases. Figure 9b shows the frequency shifts corresponding to the thicknesses. The red solid line is the result of exponential function fitting. As can be seen, as the analyte thickness increases, the frequency shift does not always increase but decreases exponentially and gradually reaches saturation after the thickness exceeds 2 μm. The outcomes are consistent with actual measurements when the thickness exceeds the saturation thickness, and the frequency shift is so weak that it can be ignored, which is also an important aspect of ensuring stable sensor measurements.

The influence of the refractive index of the analyte on the sensor performance is investigated when the thickness of the analyte is fixed at t = 2 μm. According to the results depicted in Figure 10a, the dual-frequency peaks have an obvious tendency to shift towards lower frequencies with an increase in the refractive index. The relationship between the dielectric constant and the resonant frequency is shown in Formula (4) [33]:(4)$$f_{r}=\frac{e}{\sqrt{2}\pi h}\sqrt{\frac{\mu_{c}}{\eta \varepsilon_{0}\left(\varepsilon_{r1}+\varepsilon_{r2}\right)P}}$$
where *P* is denoted as the period of the unit structure, and $\varepsilon_{r1}$ and $\varepsilon_{r2}$ are represented as the dielectric constants of different media, respectively. The trend of the frequency shift of P_1_ and P_2_ is shown in Figure 10b, which clearly shows that the frequency shift increases almost linearly with the increase in the refractive index. The curves’ steepness represents the sensor’s sensitivity, providing a more intuitive reflection of the trends in resonance peak variations. As the refractive index of the analyte changes, the absorption rate of the dual-frequency absorption peaks can always be maintained above 90%, with almost no change in amplitude. However, the phase of peak P_1_ shifts from 2.6 THz to 2.12 THz, and P_2_ shifts from 6.17 THz to 5.15 THz. Apparently, the phase shift of P_2_ is more pronounced than P_1_; this is primarily due to the different dispersion effects of the material at high or low frequencies.

To evaluate the performance of the proposed sensor quantitatively, three performance indexes are introduced, which are sensitivity (S), figure of merit (FOM) value, and quality factor (Q). The sensitivity is defined as S = Δ*f*/Δn (THz/RIU). Δ*f* denotes the frequency shift from *f*_1_ to *f*_2_, which means Δ*f* = |*f*_1_ − *f*_2_|. Δn represents the change in the refractive index from n_1_ to n_2_, that means Δn = |n_1_ − n_2_|, and the unit is THz/RIU (Refractive Index Unit, RIU). The value of FOM is a dimensionless parameter and defined as FOM = S/FWHM, where FWHM represents the full width at half maximum of the resonance peak. Q is an important parameter of the sensor. Sensors with high Q values have high sensitivity and stability, which can provide more accurate and reliable measurement results. Q is defined and expressed as Q = *f*/FWHM. For the proposed sensor, the sensitivity of P_1_ is S_1_ = 714 GHz/RIU with a quality factor of Q_1_ = 11.8 and an FOM value of 1.63 RIU^−1^. The sensitivity of P_2_ is calculated to be S_2_ = 1.627 THz/RIU, with a quality factor of Q_2_ = 29.6 and an FOM value of 3.9 RIU^−1^.

The results obtained in this paper are compared with similar works in Refs. [26,27,29,34,35,36,37,38] shown in Table 1. Compared with Refs. [29,38], in terms of pattern structure, our design is simpler and easier to fabricate. Compared with Ref. [37], the proposed sensor not only has a wider operable frequency band range, higher sensitivity, and higher Q and FOM values but can also be dynamically adjusted by the introduction of graphene. Compared with Refs. [34,35,36], the proposed sensor has dual-frequency resonance peaks and better sensing performance. Although Ref. [26] has a higher sensitivity and higher Q and FOM values, it adopts a metal structure and can only achieve the passive modulation of terahertz waves.

To illustrate the specific application of the proposed sensor, several common substances are tested, including air (RI = 1.0), tryptophan powder (RI = 1.17), water (RI = 1.33), polyethylene (PE) powder (RI = 1.5), and Benzocy clobutene (BCB, RI = 1.65). The results are shown in Figure 11. It can be observed that the resonance peak undergoes a significant redshift as the refractive index increases. The designed sensor can also be applied for practical applications in the biomedical field. Usually, biological samples attached to the sensor’s surface exhibit different dielectric constants under an external electric field, meaning that the physical properties of the surface substances can be analyzed. For example, the content of water in cells is a significant factor affecting the dielectric constant between tumor cells and healthy cells. Normal cells have a content of water of about 30–70% with a refractive index of 1.36, while basal carcinoma cells have a high content of water of 80% and a refractive index of about 1.38. Tumor cells with abundant vascularization and edema have been reported to exhibit high cellular water content. Since the vibration modes of water molecules fall in the terahertz frequency domain (the most vital resonance frequencies are 5.6 and 1.5 THz), the terahertz sensor designed in this study has essential significance in biomedicine, sample detection, and environment monitoring.

## 4. Conclusions

A dual-frequency terahertz sensor based on graphene and absorption mechanism is proposed. By analyzing the structural parameters of the top graphene layers, the optimal parameters are identified to enhance the mutual coupling between graphene and terahertz waves. The absorption rate exceeds 99% at resonant frequencies of 2.86 and 6.07 THz. The sensitivity of P_1_ is calculated to be 714 GHz/RIU, with a quality factor of Q_1_ = 11.8 and an FOM value of 1.63 RIU^−1^. The sensitivity of P_2_ was calculated to be 1.627 THz/RIU, with a quality factor of Q_2_ = 29.6 and an FOM value of 3.9 RIU^−1^. The sensing mechanism of the designed sensor is revealed by analyzing the electromagnetic field distribution. The performance of the sensor is analyzed for different refractive indices and different thicknesses of the analyte. Compared to similar research reported before, the present senor not only has a simple micro/nano-pattern but also exhibits superior sensing characteristics, such as a higher sensitivity and Q value. Meanwhile, compared to sensors made of metal, the designed sensor can be dynamically tuned. Therefore, the sensor has great potential application values in the fields of optical communication, material characterization, environmental monitoring, and biomedicine in the terahertz band.

## Figures and Tables

**Figure 1 nanomaterials-14-00378-f001:**
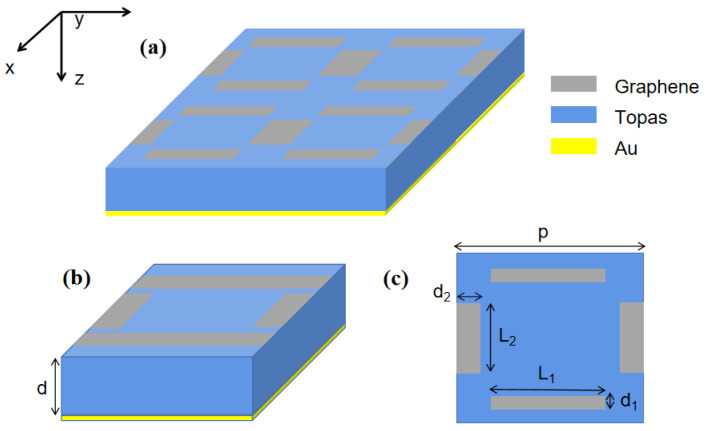
Structure of the sensor. (**a**) Schematic view; (**b**) unit cell; (**c**) top view (x-y plane).

**Figure 2 nanomaterials-14-00378-f002:**
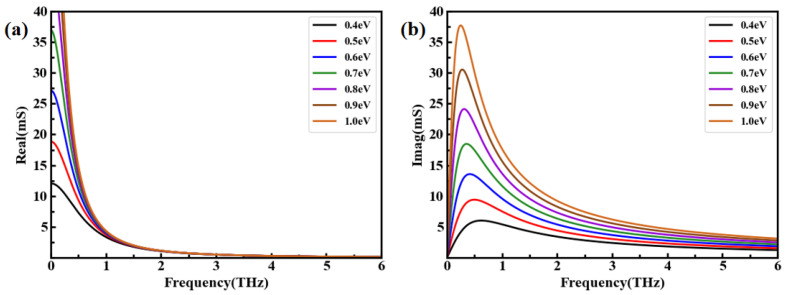
The real and imaginary parts of $\sigma_{g}$. (**a**) Real part; (**b**) imaginary part.

**Figure 3 nanomaterials-14-00378-f003:**
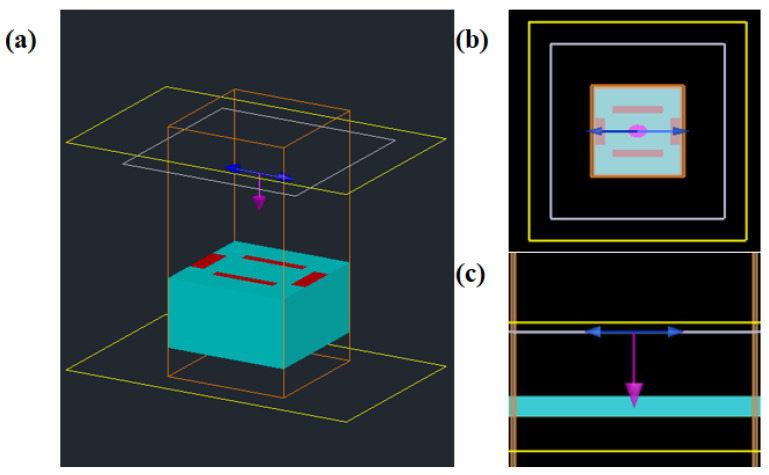
Simulation model of the sensor established in FDTD. (**a**) Perspective view of the sensor; (**b**) the view of x-y plane; (**c**) the view of y-z plane.

**Figure 4 nanomaterials-14-00378-f004:**
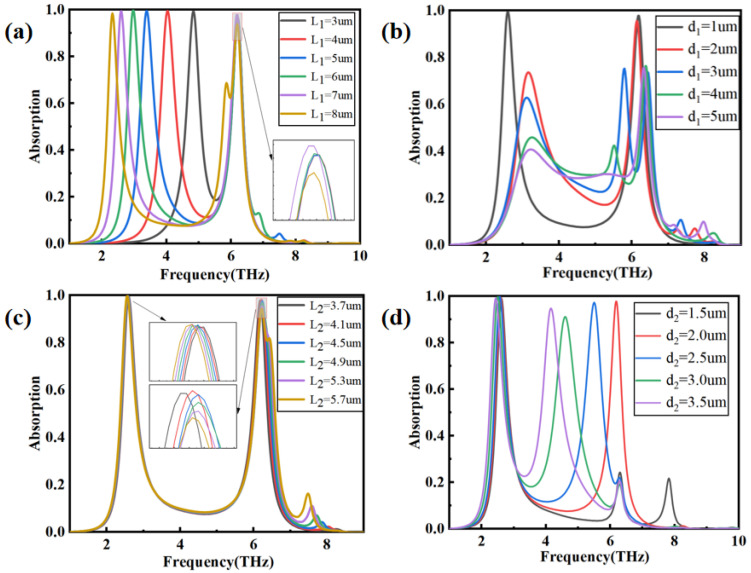
The sensing performance with different geometrical parameters in TM mode. (**a**) Different L_1_; (**b**) different d_1_; (**c**) different L_2_; (**d**) different d_2_.

**Figure 5 nanomaterials-14-00378-f005:**
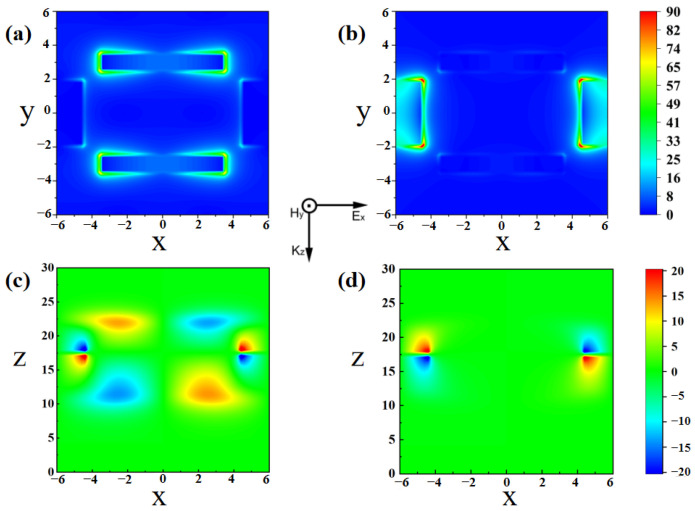
The electric field distribution at 2.58 THz and 6.07 THz, respectively. (**a**,**b**) are in the x-y plane; (**c**,**d**) are in the y-z plane.

**Figure 6 nanomaterials-14-00378-f006:**
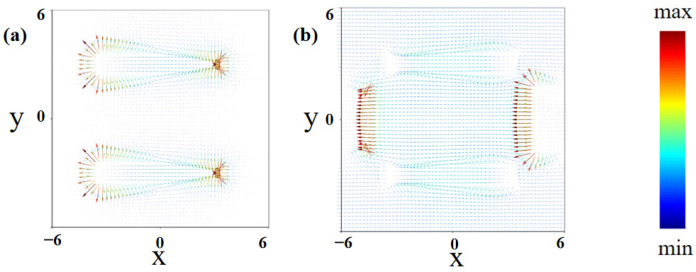
Surface current distribution in TM mode at dual-frequency peaks. (**a**) At 2.58 THz; (**b**) at 6.07 THz.

**Figure 7 nanomaterials-14-00378-f007:**
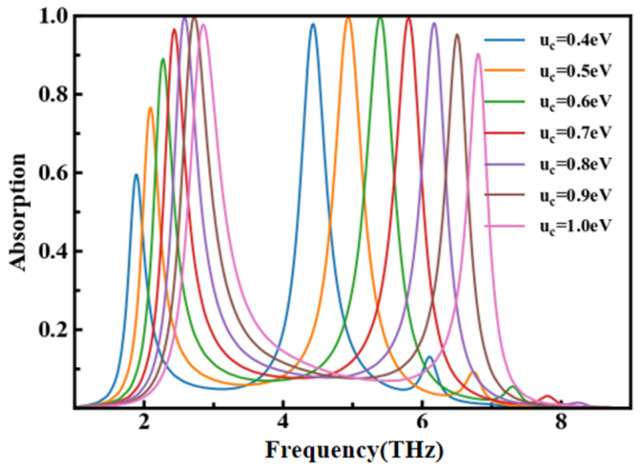
The influence of Fermi level on dual-frequency absorption peaks.

**Figure 8 nanomaterials-14-00378-f008:**
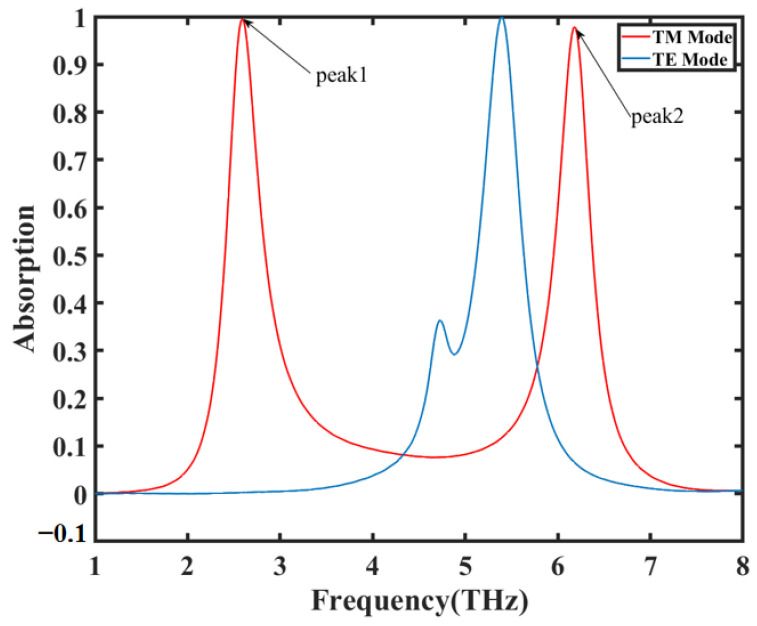
The influence of TE and TM polarization modes on dual-frequency absorption peaks.

**Figure 9 nanomaterials-14-00378-f009:**
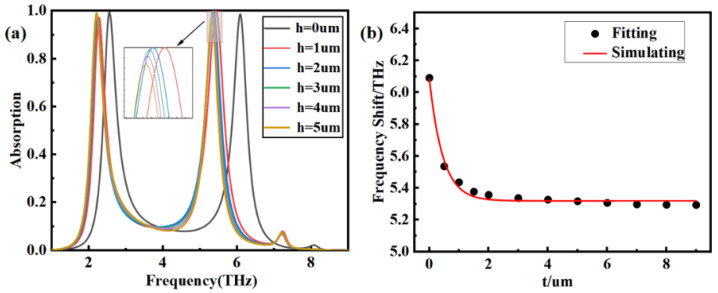
The influence of thickness of analyte on P_1_ and P_2_. (**a**) Effect of thickness on absorption spectra; (**b**) exponential function fitting curve.

**Figure 10 nanomaterials-14-00378-f010:**
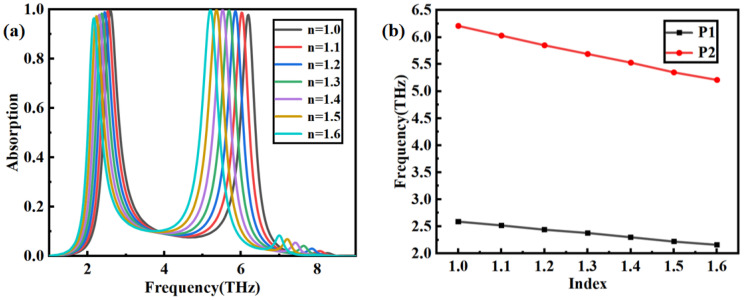
The influence of refractive index on P_1_ and P2. (**a**) Absorption spectra; (**b**) the relationship between the refractive index and the positions of P_1_ and P_2_.

**Figure 11 nanomaterials-14-00378-f011:**
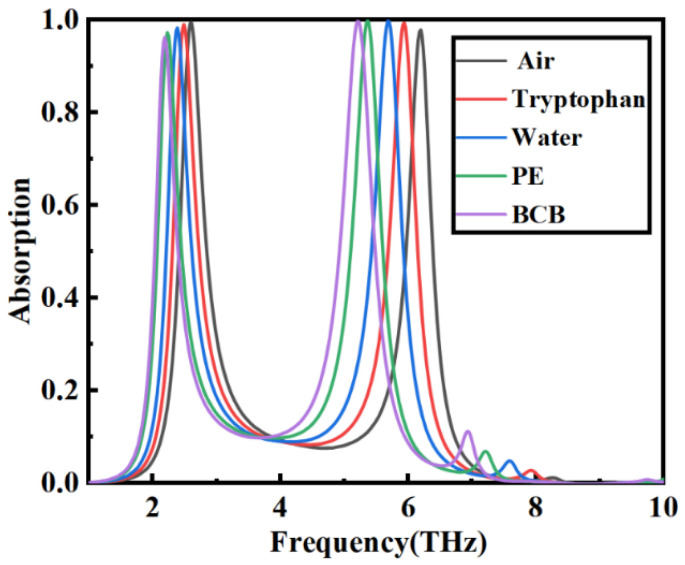
The influence of different analytes on the sensing performance.

**Table 1 nanomaterials-14-00378-t001:** Comparison of the performance of sensors from other researchers.

Ref.	Pattern	Material	Number of Peaks	Operating Range (THz)	Sensitivity (GHz/RIU)	Quality Factor (Q)	FOM (RIU^−1^)
[34]	Split ring	Gold–dielectric–gold	1	1.5–3	300	22.5	2.94
[35]	Folded Split Ring	Graphene–SiO_2_–gold	1	2–6	851	13.76	3.33
[36]	Elliptical ring	Graphene–SiO_2_	1	3–10	1326	/	3.6
[27]	cross-shaped patch	GaAs–Cu	1	2–2.5	1942	637	506
[26]	Square Au ring	Gold–dielectric–gold	1	2.5–3.5	2584	442	385
[37]	Split ring	Gold–PTFE	2	0.2–3	730	10.17	2.52
[38]	Two strips and ring	Graphene–SiO_2_	3	1–6	1083	16.2	3.78
[29]	Split ring and ribbon	Graphene–TOPAS–gold	3	2–10	1791	32.354	7.046
Our work	Four strips	Graphene–TOPAS–gold	2	1–10	1627	29.6	3.9

## Data Availability

All data that support the findings of this study are included within the article.

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
