# Peer review of "Graphene-Based Tunable Dual-Frequency Terahertz Sensor"

_nanomaterials, 2024, doi:10.3390/nano14040378_

Round 1

Reviewer 1 Report

Comments and Suggestions for Authors

The authors numerically proposed a dual-frequency terahertz sensor based on graphene. Taking into account the structural parameters of the top graphene layers, the optimal  parameters were identified to enhance the mutual coupling between the sensor structure and terahertz waves. After a thorough analysis of the manuscript, I ask the authors to provide detailed answers to the following issues:

1.     What I miss in the introduction is a thorough review of the possibilities of graphene-based metamaterials. I think it is worth mentioning a few concepts of such devices, e.g.: "Graphene-based tunable hyperbolic microcavity." Scientific Reports 11.1 (2021): 74; "Graphene-based hyperbolic metamaterial as a switchable reflection modulator." Optics Express 28.5 (2020): 6708-6718, etc.

2.     What was the thickness of the graphene monolayer in the simulation model? Did the authors use a single monolayer or several graphene monolayers? What is the impact of changing the thickness of graphene (by using several superimposed monolayers) on the device parameters? I think it is worth showing it in order to increase the scientific value of the manuscript.

3.     What was the basis for choosing a metamaterial structure consisting of 4 strip elements? What is the reason for this particular shape of the proposed structure? There are many simpler models of meta-structures that allow for greater sensitivity. Please justify this in detail.

4.     What voltage values correspond to the individual values of graphene's chemical potential?

5.     How can the absorption band be widened in the presented model?

6. I suggest inserting a table comparing the most important parameters of the presented device in relation to other THz graphene-based metamaterial sensors.

Comments on the Quality of English Language

Minor editing of English language required.

Author Response

We have carefully read the reviewers' reports on our manuscript and would like to express our sincere thanks to the reviewers and the editors for bringing us the valuable comments. We have accepted all the appropriate suggestions to improve our manuscript. We would like to describe the changes that we have made and present some explanations to respond the reviewers' comments. The changes in the manuscript are high lightened in yellow.

Reviewer 3 Report

Comments and Suggestions for Authors

Reviewer comments:

 1.      Could you elaborate more on the operational principle of the sensor? How does the proposed structure of metal bottom layer, dielectric layer, and single-layer graphene contribute to dual-band terahertz absorption?

2.      There are some grammar mistakes. I recommend checking and rereviewing the English language of this paper by a Native person.

3.      What numerical simulation methods and models were employed to obtain the absorption peaks? Providing details on the simulation parameters and methodology would enhance the clarity of the study.

4.      The absorption peaks at 2.58 THz and 6.07 THz are highlighted. What are the specific reasons for choosing these frequencies, and how do they relate to potential applications or advantages of the proposed sensor?

5.      The Q values at 2.58 THz and 6.07 THz are provided. Could you discuss the significance of these Q values in the context of terahertz sensing? How do they compare to other terahertz sensors?

6.      The tunability of the sensor is mentioned through adjustments in the external electric field or the chemical doping of graphene. Can you provide more details on the tunability mechanism and how it impacts the dual-frequency resonance peaks?

7.      The excitation of plasma resonance in graphene is mentioned as illustrating the sensor's mechanism. Could you provide more insights into how plasma resonance contributes to the sensor's functionality?

8.      The response of the sensor to different analyte samples is discussed. Could you elaborate on the types of analytes tested and the observed variations in the dual-frequency absorption peaks?

9.      The thickness of the analyte covering is mentioned as influencing the interaction between graphene and terahertz waves. Could you discuss the specific effects of varying analyte thickness on the sensor's performance?

10.   The abstract mentions potential applications in pharmaceutical detection, medical diagnosis, and environmental monitoring. Can you provide more specific examples or scenarios where this sensor could be particularly advantageous?

Best Regards,

Comments on the Quality of English Language

There are some grammar mistakes. I recommend checking and rereviewing the English language of this paper by a Native person.

Author Response

We have carefully read the reviewers' reports on our manuscript and would like to express our sincere thanks to the reviewers and the editors for bringing us the valuable comments. We have accepted all the appropriate suggestions to improve our manuscript. We would like to describe the changes that we have made and present some explanations to respond the reviewers' comments. The changes in the manuscript are high lightened in blue, green, yellow and Red font.

Round 2

Reviewer 1 Report

Comments and Suggestions for Authors

The authors responded to my suggestions and improved the manuscript. I think it can be published in its current form.

Reviewer 3 Report

Comments and Suggestions for Authors

Dear Editor of Nanomaterials,

I hope this letter finds you well. I am writing to update you on the manuscript "Graphene-based tunable dual-frequency terahertz sensor" submitted for publication in Molecules.

I am pleased to inform you that the author has diligently addressed all comments and suggestions made by referees during the peer review process. The revisions significantly improved the manuscript's overall quality.

The author's response to referee comments demonstrates a commitment to improving the paper's clarity, methodology, and overall contribution. The revisions have improved the validity and relevance of the research presented in the manuscript.

Given the author's detailed response and the significant improvements made to the manuscript, I recommend that the paper be accepted for publication in its current form. I believe it will benefit the scientific community.

Best Regards,